# $B$-tests: Low Variance Kernel Two-Sample Tests

**Wojciech Zaremba**
Center for Visual Computing
École Centrale Paris
Châtenay-Malabry, France

**Arthur Gretton**
Gatsby Unit
University College London
United Kingdom

**Matthew Blaschko**
Équipe GALEN
Inria Saclay
Châtenay-Malabry, France

{woj.zaremba,arthur.gretton}@gmail.com, matthew.blaschko@inria.fr

## Abstract

A family of maximum mean discrepancy (MMD) kernel two-sample tests is introduced. Members of the test family are called Block-tests or $B$-tests, since the test statistic is an average over MMDs computed on subsets of the samples. The choice of block size allows control over the tradeoff between test power and computation time. In this respect, the $B$-test family combines favorable properties of previously proposed MMD two-sample tests: $B$-tests are more powerful than a linear time test where blocks are just pairs of samples, yet they are more computationally efficient than a quadratic time test where a single large block incorporating all the samples is used to compute a U-statistic. A further important advantage of the $B$-tests is their asymptotically Normal null distribution: this is by contrast with the U-statistic, which is degenerate under the null hypothesis, and for which estimates of the null distribution are computationally demanding. Recent results on kernel selection for hypothesis testing transfer seamlessly to the $B$-tests, yielding a means to optimize test power via kernel choice.

## 1 Introduction

Given two samples $\{x_i\}_{i=1}^n$ where $x_i \sim P$ i.i.d., and $\{y_i\}_{i=1}^n$, where $y_i \sim Q$ i.i.d, the two sample problem consists in testing whether to accept or reject the null hypothesis $\mathcal{H}_0$ that $P = Q$, vs the alternative hypothesis $\mathcal{H}_A$ that $P$ and $Q$ are different. This problem has recently been addressed using measures of similarity computed in a reproducing kernel Hilbert space (RKHS), which apply in very general settings where $P$ and $Q$ might be distributions over high dimensional data or structured objects. Kernel test statistics include the maximum mean discrepancy [10, 6] (of which the energy distance is an example [18, 2, 22]), which is the distance between expected features of $P$ and $Q$ in the RKHS; the kernel Fisher discriminant [12], which is the distance between expected feature maps normalized by the feature space covariance; and density ratio estimates [24]. When used in testing, it is necessary to determine whether the empirical estimate of the relevant similarity measure is sufficiently large as to give the hypothesis $P = Q$ low probability; i.e., below a user-defined threshold $\alpha$, denoted the test level. The test power denotes the probability of correctly rejecting the null hypothesis, given that $P \neq Q$.

The minimum variance unbiased estimator $\mathrm{MMD}_u$ of the maximum mean discrepancy, on the basis of $n$ samples observed from each of $P$ and $Q$, is a U-statistic, costing $O(n^2)$ to compute. Unfortunately, this statistic is degenerate under the null hypothesis $\mathcal{H}_0$ that $P = Q$, and its asymptotic distribution takes the form of an infinite weighted sum of independent $\chi^2$ variables (it is asymptotically Gaussian under the alternative hypothesis $\mathcal{H}_A$ that $P \neq Q$). Two methods for empirically estimating the null distribution in a consistent way have been proposed: the bootstrap [10], and a method requiring an eigendecomposition of the kernel matrices computed on the merged samples from $P$ and $Q$ [7]. Unfortunately, both procedures are computationally demanding: the former costs $O(n^2)$, with a large constant (the MMD must be computed repeatedly over random assignments of the pooled data); the latter costs $O(n^3)$, but with a smaller constant, hence can in practice be

faster than the bootstrap. Another approach is to approximate the null distribution by a member of a simpler parametric family (for instance, a Pearson curve approximation), however this has no consistency guarantees.

More recently, an $O(n)$ unbiased estimate $\mathrm{MMD}_l$ of the maximum mean discrepancy has been proposed [10, Section 6], which is simply a running average over independent pairs of samples from $P$ and $Q$. While this has much greater variance than the U-statistic, it also has a simpler null distribution: being an average over i.i.d. terms, the central limit theorem gives an asymptotically Normal distribution, under both $\mathcal{H}_0$ and $\mathcal{H}_A$. It is shown in [9] that this simple asymptotic distribution makes it easy to optimize the Hodges and Lehmann asymptotic relative efficiency [19] over the family of kernels that define the statistic: in other words, to choose the kernel which gives the lowest Type II error (probability of wrongly accepting $\mathcal{H}_0$) for a given Type I error (probability of wrongly rejecting $\mathcal{H}_0$). Kernel selection for the U-statistic is a much harder question due to the complex form of the null distribution, and remains an open problem.

It appears that $\mathrm{MMD}_u$ and $\mathrm{MMD}_l$ fall at two extremes of a spectrum: the former has the lowest variance of any $n$-sample estimator, and should be used in limited data regimes; the latter is the estimator requiring the least computation while still looking at each of the samples, and usually achieves better Type II error than $\mathrm{MMD}_u$ at a given computational cost, albeit by looking at much more data (the "limited time, unlimited data" scenario). A major reason $\mathrm{MMD}_l$ is faster is that its null distribution is straightforward to compute, since it is Gaussian and its variance can be calculated at the same cost as the test statistic. A reasonable next step would be to find a compromise between these two extremes: to construct a statistic with a lower variance than $\mathrm{MMD}_l$, while retaining an asymptotically Gaussian null distribution (hence remaining faster than tests based on $\mathrm{MMD}_u$). We study a family of such test statistics, where we split the data into blocks of size $B$, compute the quadratic-time $\mathrm{MMD}_u$ on each block, and then average the resulting statistics. We call the resulting tests $B$-tests. As long as we choose the size $B$ of blocks such that $n/B \to \infty$, we are still guaranteed asymptotic Normality by the central limit theorem, and the null distribution can be computed at the same cost as the test statistic. For a given sample size $n$, however, the power of the test can increase dramatically over the $\mathrm{MMD}_l$ test, even for moderate block sizes $B$, making much better use of the available data with only a small increase in computation.

The block averaging scheme was originally proposed in [13], as an instance of a two-stage U-statistic, to be applied when the degree of degeneracy of the U-statistic is indeterminate. Differences with respect to our method are that Ho and Shieh compute the block statistics by sampling with replacement [13, (b) p. 863], and propose to obtain the variance of the test statistic via Monte Carlo, jackknife, or bootstrap techniques, whereas we use closed form expressions. Ho and Shieh further suggest an alternative two-stage U-statistic in the event that the degree of degeneracy is known; we return to this point in the discussion. While we confine ourselves to the MMD in this paper, we emphasize that the block approach applies to a much broader variety of test situations where the null distribution cannot easily be computed, including the energy distance and distance covariance [18, 2, 22] and Fisher statistic [12] in the case of two-sample testing, and the Hilbert-Schmidt Independence Criterion [8] and distance covariance [23] for independence testing. Finally, the kernel learning approach of [9] applies straightforwardly, allowing us to maximize test power over a given kernel family. Code is available at http://github.com/wojzaremba/btest.

## 2 Theory

In this section we describe the mathematical foundations of the $B$-test. We begin with a brief review of kernel methods, and of the maximum mean discrepancy. We then present our block-based average MMD statistic, and derive its distribution under the $\mathcal{H}_0$ ($P = Q$) and $\mathcal{H}_A$ ($P \neq Q$) hypotheses. The central idea employed in the construction of the $B$-test is to generate a low variance MMD estimate by averaging multiple low variance kernel statistics computed over blocks of samples. We show simple sufficient conditions on the block size for consistency of the estimator. Furthermore, we analyze the properties of the finite sample estimate, and propose a consistent strategy for setting the block size as a function of the number of samples.

### 2.1 Definition and asymptotics of the block-MMD

Let $\mathcal{F}_k$ be an RKHS defined on a topological space $\mathcal{X}$ with reproducing kernel $k$, and $P$ a Borel probability measure on $\mathcal{X}$. The *mean embedding* of $P$ in $\mathcal{F}_k$, written $\mu_k(p) \in \mathcal{F}_k$ is defined such

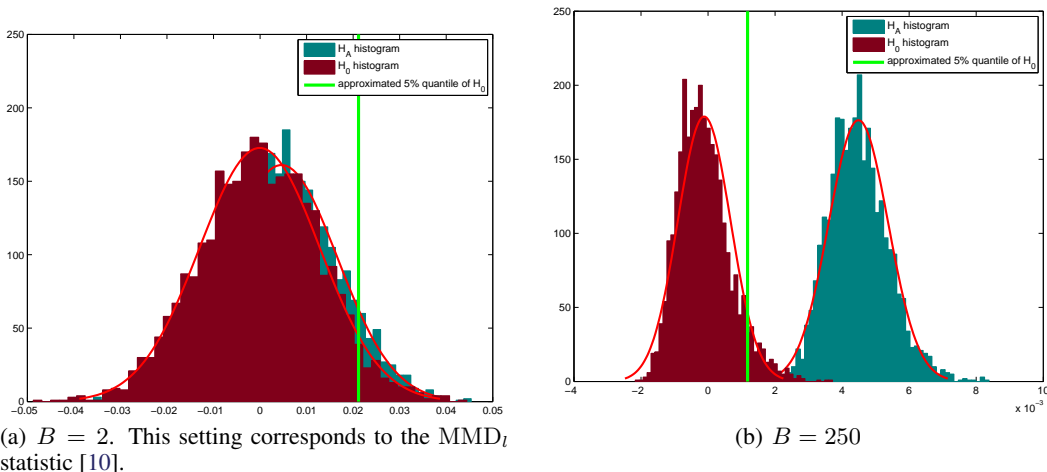

(a) $B = 2$. This setting corresponds to the $\mathrm{MMD}_l$ statistic [10].

(b) $B = 250$

Figure 1: Empirical distributions under $\mathcal{H}_0$ and $\mathcal{H}_A$ for different regimes of $B$ for the music experiment (Section 3.2). In both plots, the number of samples is fixed at 500. As we vary $B$, we trade off the quality of the finite sample Gaussian approximation to the null distribution, as in Theorem 2.3, with the variances of the $\mathcal{H}_0$ and $\mathcal{H}_A$ distributions, as outlined in Section 2.1. In (b) the distribution under $\mathcal{H}_0$ does not resemble a Gaussian (it does not pass a level 0.05 Kolmogorov-Smirnov (KS) normality test [16, 20]), and a Gaussian approximation results in a conservative test threshold (vertical green line). The remaining empirical distributions all pass a KS normality test.

that $E_{x \sim p} f(x) = \langle f, \mu_k(p) \rangle_{\mathcal{F}_k}$ for all $f \in \mathcal{F}_k$, and exists for all Borel probability measures when $k$ is bounded and continuous [3, 10]. The maximum mean discrepancy (MMD) between a Borel probability measure $P$ and a second Borel probability measure $Q$ is the squared RKHS distance between their respective mean embeddings,

$$\eta_k(P, Q) = \|\mu_k(P) - \mu_k(Q)\|_{\mathcal{F}_k}^2 = E_{xx'} k(x, x') + E_{yy'} k(y, y') - 2E_{xy} k(x, y), \qquad (1)$$

where $x'$ denotes an independent copy of $x$ [11]. Introducing the notation $z = (x, y)$, we write

$$\eta_k(P, Q) = E_{zz'} h_k(z, z'), \qquad h(z, z') = k(x, x') + k(y, y') - k(x, y') - k(x', y). \qquad (2)$$

When the kernel $k$ is characteristic, then $\eta_k(P, Q) = 0$ iff $P = Q$ [21]. Clearly, the minimum variance unbiased estimate $\mathrm{MMD}_u$ of $\eta_k(P, Q)$ is a U-statistic.

By analogy with $\mathrm{MMD}_u$, we make use of averages of $h(x, y, x', y')$ to construct our two-sample test. We denote by $\hat{\eta}_k(i)$ the $i$th empirical estimate $\mathrm{MMD}_u$ based on a subsample of size $B$, where $1 \le i \le \frac{n}{B}$ (for notational purposes, we will index samples as though they are presented in a random fixed order). More precisely,

$$\hat{\eta}_k(i) = \frac{1}{B(B-1)} \sum_{a=(i-1)B+1}^{iB} \sum_{b=(i-1)B+1, b \ne a}^{iB} h(z_a, z_b). \qquad (3)$$

The $B$-test statistic is an MMD estimate obtained by averaging the $\hat{\eta}_k(i)$. Each $\hat{\eta}_k(i)$ under $\mathcal{H}_0$ converges to an infinite sum of weighted $\chi^2$ variables [7]. Although setting $B = n$ would lead to the lowest variance estimate of the MMD, computing sound thresholds for a given $p$-value is expensive, involving repeated bootstrap sampling [5, 14], or computing the eigenvalues of a Gram matrix [7].

In contrast, we note that $\hat{\eta}_k(i)_{i=1,\ldots,\frac{n}{B}}$ are i.i.d. variables, and averaging them allows us to apply the central limit theorem in order to estimate $p$-values from a normal distribution. We denote the average of the $\hat{\eta}_k(i)$ by $\hat{\eta}_k$,

$$\hat{\eta}_k = \frac{B}{n} \sum_{i=1}^{\frac{n}{B}} \hat{\eta}_k(i). \qquad (4)$$

We would like to apply the central limit theorem to variables $\hat{\eta}_k(i)_{i=1,\ldots,\frac{n}{B}}$. It remains for us to derive the distribution of $\hat{\eta}_k$ under $\mathcal{H}_0$ and under $\mathcal{H}_A$. We rely on the result from [11, Theorem 8] for $\mathcal{H}_A$. According to our notation, for every $i$,

**Theorem 2.1** *Assume $0 < \mathbb{E}(h^2) < \infty$, then under $\mathcal{H}_A$, $\hat{\eta}_k$ converges in distribution to a Gaussian according to*

$$B^{\frac{1}{2}}(\hat{\eta}_k(i) - \text{MMD}^2) \xrightarrow{D} \mathcal{N}(0, \sigma_u^2), \tag{5}$$

*where $\sigma_u^2 = 4\left(\mathbb{E}_z[(\mathbb{E}_{z'}h(z, z'))^2 - \mathbb{E}_{z,z'}(h(z, z'))]^2\right)$.*

This in turn implies that

$$\hat{\eta}_k(i) \xrightarrow{D} \mathcal{N}(\text{MMD}^2, \sigma_u^2 B^{-1}). \tag{6}$$

For an average of $\{\hat{\eta}_k(i)\}_{i=1,\ldots,\frac{n}{B}}$, the central limit theorem implies that under $\mathcal{H}_A$,

$$\hat{\eta}_k \xrightarrow{D} \mathcal{N}\left(\text{MMD}^2, \sigma_u^2\left(Bn/B\right)^{-1}\right) = \mathcal{N}\left(\text{MMD}^2, \sigma_u^2 n^{-1}\right). \tag{7}$$

This result shows that the distribution of $\mathcal{H}_A$ is asymptotically independent of the block size, $B$. Turning to the null hypothesis, [11, Theorem 8] additionally implies that under $\mathcal{H}_0$ for every $i$,

**Theorem 2.2**

$$B\hat{\eta}_k(i) \xrightarrow{D} \sum_{l=1}^{\infty} \lambda_l[z_l^2 - 2], \tag{8}$$

*where $z_l \sim \mathcal{N}(0, 2)^2$ i.i.d, $\lambda_l$ are the solutions to the eigenvalue equation*

$$\int_{\mathcal{X}} \bar{k}(x, x')\psi_l(x)dp(x) = \lambda_l\psi_l(x'), \tag{9}$$

*and $\bar{k}(x_i, x_j) := k(x_i, x_j) - \mathbb{E}_x k(x_i, x) - \mathbb{E}_x k(x, x_j) + \mathbb{E}_{x,x'} k(x, x')$ is the centered RKHS kernel.*

As a consequence, under $\mathcal{H}_0$, $\hat{\eta}_k(i)$ has expected variance $2B^{-2}\sum_{l=1}^{\infty} \lambda^2$. We will denote this variance by $CB^{-2}$. The central limit theorem implies that under $\mathcal{H}_0$,

$$\hat{\eta}_k \xrightarrow{D} \mathcal{N}\left(0, C\left(B^2 n/B\right)^{-1}\right) = \mathcal{N}\left(0, C(nB)^{-1}\right) \tag{10}$$

The asymptotic distributions for $\hat{\eta}_k$ under $\mathcal{H}_0$ and $\mathcal{H}_A$ are Gaussian, and consequently it is easy to calculate the distribution quantiles and test thresholds. Asymptotically, it is always beneficial to increase $B$, as the distributions for $\eta$ under $\mathcal{H}_0$ and $\mathcal{H}_A$ will be better separated. For consistency, it is sufficient to ensure that $n/B \to \infty$.

A related strategy of averaging over data blocks to deal with large sample sizes has recently been developed in [15], with the goal of efficiently computing bootstrapped estimates of statistics of interest (e.g. quantiles or biases). Briefly, the approach splits the data (of size $n$) into $s$ subsamples each of size $B$, computes an estimate of the $n$-fold bootstrap on each block, and averages these estimates. The difference with respect to our approach is that we use the asymptotic distribution of the average over block statistics to determine a threshold for a hypothesis test, whereas [15] is concerned with proving the consistency of a statistic obtained by averaging over bootstrap estimates on blocks.

## 2.2 Convergence of Moments

In this section, we analyze the convergence of the moments of the $B$-test statistic, and comment on potential sources of bias.

The central limit theorem implies that the empirical mean of $\{\hat{\eta}_k(i)\}_{i=1,\ldots,\frac{n}{B}}$ converges to $\mathbb{E}(\hat{\eta}_k(i))$. Moreover it states that the variance $\{\hat{\eta}_k(i)\}_{i=1,\ldots,\frac{n}{B}}$ converges to $\mathbb{E}(\hat{\eta}_k(i))^2 - \mathbb{E}(\hat{\eta}_k(i)^2)$. Finally, all remaining moments tend to zero, where the rate of convergence for the $j$th moment is of the order $\left(\frac{n}{B}\right)^{\frac{j+1}{2}}$ [1]. This indicates that the skewness dominates the difference of the distribution from a Gaussian.

Under both $\mathcal{H}_0$ and $\mathcal{H}_A$, thresholds computed from normal distribution tables are asymptotically unbiased. For finite samples sizes, however, the bias under $\mathcal{H}_0$ can be more severe. From Equation (8) we have that under $\mathcal{H}_0$, the summands, $\hat{\eta}_k(i)$, converge in distribution to infinite weighted sums of $\chi^2$ distributions. Every unweighted term of this infinite sum has distribution $\mathcal{N}(0, 2)^2$, which has finite skewness equal to $8$. The skewness for the entire sum is finite and positive,

$$C = \sum_{l=1}^{\infty} 8\lambda_l^3, \tag{11}$$

as $\lambda_l \geq 0$ for all $l$ due to the positive definiteness of the kernel $k$. The skew for the mean of the $\hat{\eta}_k(i)$ converges to 0 and is positively biased. At smaller sample sizes, test thresholds obtained from the standard Normal table may therefore be inaccurate, as they do not account for this skew. In our experiments, this bias caused the tests to be overly conservative, with lower Type I error than the design level required (Figures 2 and 5).

## 2.3 Finite Sample Case

In the finite sample case, we apply the Berry-Esséen theorem, which gives conservative bounds on the $\ell_\infty$ convergence of a series of finite sample random variables to a Gaussian distribution [4].

**Theorem 2.3** *Let* $X_1, X_2, \ldots, X_n$ *be i.i.d. variables.* $\mathbb{E}(X_1) = 0$, $\mathbb{E}(X_1^2) = \sigma^2 > 0$, *and* $\mathbb{E}(|X_1|^3) = \rho < \infty$. *Let* $F_n$ *be a cumulative distribution of* $\frac{\sum_{i=1}^{n} X_i}{\sqrt{n}\sigma}$, *and let* $\Phi$ *denote the standard normal distribution. Then for every* $x$,

$$|F_n(x) - \Phi(x)| \leq C\rho\sigma^{-3}n^{-1/2}, \tag{12}$$

*where* $C < 1$.

This result allows us to ensure fast point-wise convergence of the $B$-test. We have that $\rho(\hat{\eta}_k) = O(1)$, i.e., it is dependent only on the underlying distributions of the samples and not on the sample size. The number of i.i.d. samples is $nB^{-1}$. Based on Theorem 2.3, the point-wise error can be upper bounded by $\frac{O(1)}{O(B^{-1})^{\frac{3}{2}}\sqrt{\frac{n}{B}}} = O(\frac{B^2}{\sqrt{n}})$ under $\mathcal{H}_A$. Under $\mathcal{H}_0$, the error can be bounded by $\frac{O(1)}{O(B^{-2})^{\frac{3}{2}}\sqrt{\frac{n}{B}}} = O(\frac{B^{3.5}}{\sqrt{n}})$.

While the asymptotic results indicate that convergence to an optimal predictor is fastest for larger $B$, the finite sample results support decreasing the size of $B$ in order to have a sufficient number of samples for application of the central limit theorem. As long as $B \to \infty$ and $\frac{n}{B} \to \infty$, the assumptions of the $B$-test are fulfilled.

By varying $B$, we make a fundamental tradeoff in the construction of our two sample test. When $B$ is small, we have many samples, hence the null distribution is close to the asymptotic limit provided by the central limit theorem, and the Type I error is estimated accurately. The disadvantage of a small $B$ is a lower test power for a given sample size. Conversely, if we increase $B$, we will have a lower variance empirical distribution for $\mathcal{H}_0$, hence higher test power, but we may have a poor estimate of the number of Type I errors (Figure 1). A sensible family of heuristics therefore is to set

$$B = [n^\gamma] \tag{13}$$

for some $0 < \gamma < 1$, where we round to the nearest integer. In this setting the number of samples available for application of the central limit theorem will be $[n^{(1-\gamma)}]$. For given $\gamma$ computational complexity of the $B$-test is $O(n^{1+\gamma})$. We note that any value of $\gamma \in (0, 1)$ yields a consistent estimator. We have chosen $\gamma = \frac{1}{2}$ in the experimental results section, with resulting complexity $O(n^{1.5})$: we emphasize that this is a heuristic, and just one choice that fulfils our assumptions.

## 3 Experiments

We have conducted experiments on challenging synthetic and real datasets in order to empirically measure (i) sample complexity, (ii) computation time, and (iii) Type I / Type II errors. We evaluate $B$-test performance in comparison to the $\mathrm{MMD}_l$ and $\mathrm{MMD}_u$ estimators, where for the latter we compare across different strategies for null distribution quantile estimation.

| Method | Kernel parameters | Additional parameters | Minimum number of samples | Computation time (s) | Consistent |
|---|---|---|---|---|---|
| | $\sigma = 1$ | $B = 2$ | 26400 | 0.0012 | ✓ |
| | | $B = 8$ | 3850 | 0.0039 | ✓ |
| | | $B = \sqrt{n}$ | 886 | 0.0572 | ✓ |
| $B$-test | $\sigma = $ median | any $B$ | $> 60000$ | | ✓ |
| | multiple kernels | $B = 2$ | 37000 | 0.0700 | ✓ |
| | | $B = 8$ | 5400 | 0.1295 | ✓ |
| | | $B = \sqrt{\frac{n}{2}}$ | 1700 | 0.8332 | ✓ |
| Pearson curves | $\sigma = 1$ | | 186 | 387.4649 | ✗ |
| Gamma approximation | | | 183 | 0.2667 | ✗ |
| Gram matrix spectrum | | | 186 | 407.3447 | ✓ |
| Bootstrap | | $B = n$ | 190 | 129.4094 | ✓ |
| Pearson curves | $\sigma = $ median | | $> 60000$, or 2h per iteration timeout | | ✗ |
| Gamma approximation | | | | | ✗ |
| Gram matrix spectrum | | | | | ✓ |
| Bootstrap | | | | | ✓ |

Table 1: Sample complexity for tests on the distributions described in Figure 3. The fourth column indicates the minimum number of samples necessary to achieve Type I and Type II errors of 5%. The fifth column is the computation time required for 2000 samples, and is not presented for settings that have unsatisfactory sample complexity.

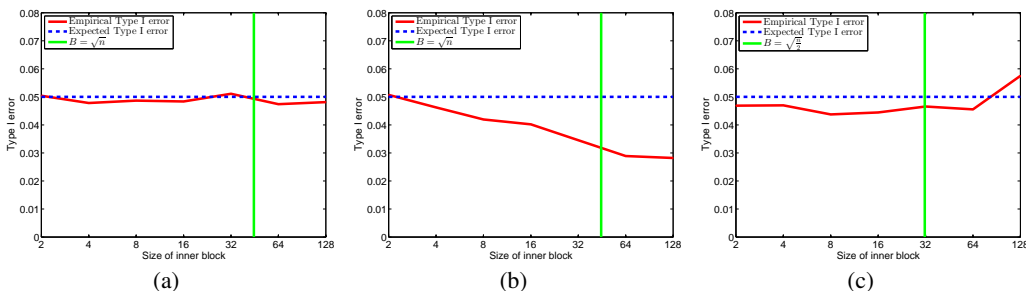

(a)      (b)      (c)

Figure 2: Type I errors on the distributions shown in Figure 3 for $\alpha = 5\%$: (a) MMD, single kernel, $\sigma = 1$, (b) MMD, single kernel, $\sigma$ set to the median pairwise distance, and (c) MMD, non-negative linear combination of multiple kernels. The experiment was repeated 30000 times. Error bars are not visible at this scale.

## 3.1 Synthetic data

Following previous work on kernel hypothesis testing [9], our synthetic distributions are $5 \times 5$ grids of 2D Gaussians. We specify two distributions, $P$ and $Q$. For distribution $P$ each Gaussian has identity covariance matrix, while for distribution $Q$ the covariance is non-spherical. Samples drawn from $P$ and $Q$ are presented in Figure 3. These distributions have proved to be very challenging for existing non-parametric two-sample tests [9].

We employed three different kernel selection strategies in the hypothesis test. First, we used a Gaussian kernel with $\sigma = 1$, which approximately matches the scale of the variance of each Gaussian in mixture $P$. While this is a somewhat arbitrary default choice, we selected it as it performs well in practice (given the lengthscale of the data), and we treat it as a baseline. Next, we set $\sigma$ equal to the median pairwise distance over the training data, which is a standard way to choose the Gaussian kernel bandwidth [17], although it is likewise arbitrary in this context. Finally, we applied a kernel learning strategy, in



(a) Distribution $P$      (b) Distribution $Q$

Figure 3: Synthetic data distributions $P$ and $Q$. Samples belonging to these classes are difficult to distinguish.

which the kernel was optimized to maximize the test power for the alternative $P \neq Q$ [9]. This approach returned a non-negative linear combination combination of base kernels, where half the data were used in learning the kernel weights (these data were excluded from the testing phase).

The base kernels in our experiments were chosen to be Gaussian, with bandwidths in the set $\sigma \in \{2^{-15}, 2^{-14}, \ldots, 2^{10}\}$. Testing was conducted using the remaining half of the data.

For comparison with the quadratic time $U$-statistic $\mathrm{MMD}_u$ [7, 10], we evaluated four null distribution estimates: (i) Pearson curves, (ii) gamma approximation, (iii) Gram matrix spectrum, and (iv) bootstrap. For methods using Pearson curves and the Gram matrix spectrum, we drew 500 samples from the null distribution estimates to obtain the $1 - \alpha$ quantiles, for a test of level $\alpha$. For the bootstrap, we fixed the number of shuffles to 1000. We note that Pearson curves and the gamma approximation are not statistically consistent. We considered only the setting with $\sigma = 1$ and $\sigma$ set to the median pairwise distance, as kernel selection is not yet solved for tests using $\mathrm{MMD}_u$ [9].

In the first experiment we set the Type I error to be 5%, and we recorded the Type II error. We conducted these experiments on 2000 samples over 1000 repetitions, with varying block size, $B$. Figure 4 presents results for different kernel choice strategies, as a function of $B$. The median heuristic performs extremely poorly in this

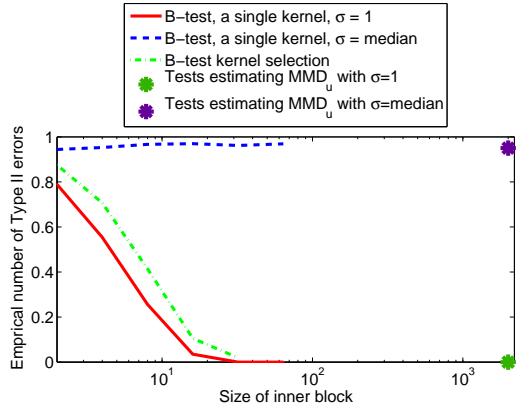

Figure 4: Synthetic experiment: number of Type II errors vs $B$, given a fixed probability $\alpha$ of Type I errors. As $B$ grows, the Type II error drops quickly when the kernel is appropriately chosen. The kernel selection method is described in [9], and closely approximates the baseline performance of the well-informed user choice of $\sigma = 1$.

experiment. As discussed in [9, Section 5], the reason for this failure is that the lengthscale of the difference between the distributions $P$ and $Q$ differs from the lengthscale of the main data variation as captured by the median, which gives too broad a kernel for the data.

In the second experiment, our aim was to compare the empirical sample complexity of the various methods. We again fixed the same Type I error for all methods, but this time we also fixed a Type II error of 5%, increasing the number of samples until the latter error rate was achieved. Column four of Table 1 shows the number of samples required in each setting to achieve these error rates. We additionally compared the computational efficiency of the various methods. The computation time for each method with a fixed sample size of 2000 is presented in column five of Table 1. All experiments were run on a single 2.4 GHz core.

Finally, we evaluated the empirical Type I error for $\alpha = 5\%$ and increasing $B$. Figure 2 displays the empirical Type I error, where we note the location of the $\gamma = 0.5$ heuristic in Equation (13). For the user-chosen kernel ($\sigma = 1$, Figure 2(a)), the number of Type I errors closely matches the targeted test level. When median heuristic is used, however, the test is overly conservative, and makes fewer Type I errors than required (Figure 2(b)). This indicates that for this choice of $\sigma$, we are not in the asymptotic regime, and our Gaussian null distribution approximation is inaccurate. Kernel selection via the strategy of [9] alleviates this problem (Figure 2(c)). This setting coincides with a block size substantially larger than 2 ($\mathrm{MMD}_l$), and therefore achieves lower Type II errors while retaining the targeted Type I error.

## 3.2 Musical experiments

In this set of experiments, two amplitude modulated Rammstein songs were compared (*Sehnsucht* vs. *Engel*, from the album *Sehnsucht*). Following the experimental setting in [9, Section 5], samples from $P$ and $Q$ were extracts from AM signals of time duration $8.3 \times 10^{-3}$ seconds in the original audio. Feature extraction was identical to [9], except that the amplitude scaling parameter was set to 0.3 instead of 0.5. As the feature vector had size 1000 we set the block size $B = \lceil \sqrt{1000} \rceil = 32$. Table 2 summarizes the empirical Type I and Type II errors over 1000 repetitions, and the average computation times. Figure 5 shows the average number of Type I errors as a function of $B$: in this case, all kernel selection strategies result in conservative tests (lower Type I error than required), indicating that more samples are needed to reach the asymptotic regime. Figure 1 shows the empirical $\mathcal{H}_0$ and $\mathcal{H}_A$ distributions for different $B$.

## 4 Discussion

We have presented experimental results both on a difficult synthetic problem, and on real-world data from amplitude modulated audio recordings. The results show that the $B$-test has a much better

| Method | Kernel parameters | Additional parameters | Type I error | Type II error | Computational time (s) |
|---|---|---|---|---|---|
| $B$-test | $\sigma = 1$ | $B = 2$ | 0.038 | 0.927 | 0.039 |
| | | $B = \sqrt{n}$ | 0.006 | 0.597 | 1.276 |
| | $\sigma = $ median | $B = 2$ | 0.043 | 0.786 | 0.047 |
| | | $B = \sqrt{n}$ | 0.026 | 0 | 1.259 |
| | multiple kernels | $B = 2$ | 0.0481 | 0.867 | 0.607 |
| | | $B = \sqrt{\frac{n}{2}}$ | 0.025 | 0.012 | 18.285 |
| Gram matrix spectrum Bootstrap | $\sigma = 1$ | $B = 2000$ | 0 | 0 | 160.1356 |
| | | | 0.01 | 0 | 121.2570 |
| Gram matrix spectrum Bootstrap | $\sigma = $ median | | 0 | 0 | 286.8649 |
| | | | 0.01 | 0 | 122.8297 |

Table 2: A comparison of consistent tests on the music experiment described in Section 3.2. Here computation time is reported for the test achieving the stated error rates.

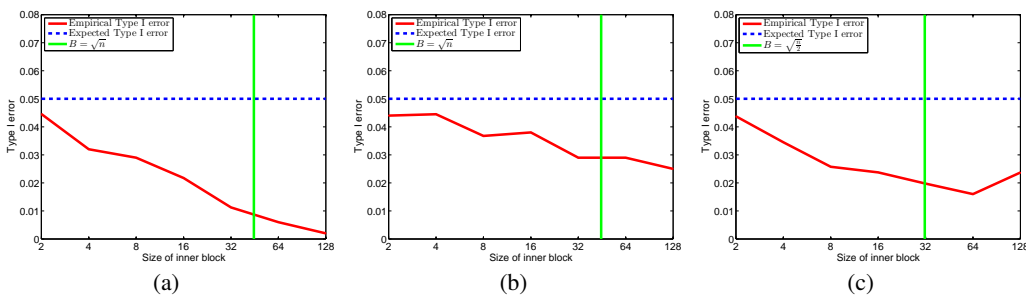

Figure 5: Empirical Type I error rate for $\alpha = 5\%$ on the music data (Section 3.2). (a) A single kernel test with $\sigma = 1$, (b) A single kernel test with $\sigma = $ median, and (c) for multiple kernels. Error bars are not visible at this scale. The results broadly follow the trend visible from the synthetic experiments.

sample complexity than $\mathrm{MMD}_l$ over all tested kernel selection strategies. Moreover, it is an order of magnitude faster than any test that consistently estimates the null distribution for $\mathrm{MMD}_u$ (i.e., the Gram matrix eigenspectrum and bootstrap estimates): these estimates are impractical at large sample sizes, due to their computational complexity. Additionally, the $B$-test remains statistically consistent, with the best convergence rates achieved for large $B$. The $B$-test combines the best features of $\mathrm{MMD}_l$ and $\mathrm{MMD}_u$ based two-sample tests: consistency, high statistical efficiency, and high computational efficiency.

A number of further interesting experimental trends may be seen in these results. First, we have observed that the empirical Type I error rate is often conservative, and is less than the $5\%$ targeted by the threshold based on a Gaussian null distribution assumption (Figures 2 and 5). In spite of this conservatism, the Type II performance remains strong (Tables 1 and 2), as the gains in statistical power of the $B$-tests improve the testing performance (cf. Figure 1). Equation (7) implies that the size of $B$ does not influence the asymptotic variance under $\mathcal{H}_A$, however we observe in Figure 1 that the empirical variance of $\mathcal{H}_A$ drops with larger $B$. This is because, for these $P$ and $Q$ and small $B$, the null and alternative distributions have considerable overlap. Hence, given the distributions are effectively indistinguishable at these sample sizes $n$, the variance of the alternative distribution as a function of $B$ behaves more like that of $\mathcal{H}_0$ (cf. Equation (10)). This effect will vanish as $n$ grows.

Finally, [13] propose an alternative approach for U-statistic based testing when the degree of degeneracy is known: a new U-statistic (the TU-statistic) is written in terms of products of centred U-statistics computed on the individual blocks, and a test is formulated using this TU-statistic. Ho and Shieh show that a TU-statistic based test can be asymptotically more powerful than a test using a single U-statistic on the whole sample, when the latter is degenerate under $\mathcal{H}_0$, and nondegenerate under $\mathcal{H}_A$. It is of interest to apply this technique to MMD-based two-sample testing.

**Acknowledgments** We thank Mladen Kolar for helpful discussions. This work is partially funded by ERC Grant 259112, and by the Royal Academy of Engineering through the Newton Alumni Scheme.

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
