[Supplementary Material]

# $B$-tests: Low Variance Kernel Two-Sample Tests–Supplementary Material

**Wojciech Zaremba**
Center for Visual Computing
École Centrale Paris
Châtenay-Malabry, France

**Arthur Gretton**
Gatsby Unit
University College London
United Kingdom

**Matthew Blaschko**
Équipe GALEN
Inria Saclay
Châtenay-Malabry, France

{woj.zaremba,arthur.gretton}@gmail.com, matthew.blaschko@inria.fr

## 1 Experiments

### 1.1 Detecting simple differences in three synthetic benchmarks

In this additional set of experiments, the data are as described in Sec. B.1 of [1]. Results are shown in Table 1 and Figure 1.

| Dataset | Method | Parameters | Type I error | Type II error | Computational time (s) |
|---|---|---|---|---|---|
| difference of means | $B$-test | $B = 2$ <br> $B = \sqrt{n}$ | 0.056 <br> 0.037 | 0.384 <br> 0 | 0.0040 <br> 0.1912 |
| | Gram matrix spectrum Bootstrap | $B = 2000$ | 0.07 <br> 0.05 | 0 <br> 0 | 511.801 <br> 122.399 |
| difference of variances | $B$-test | $B = 2$ <br> $B = \sqrt{n}$ | 0.056 <br> 0.037 | 0.337 <br> 0 | 0.0040 <br> 0.2043 |
| | Gram matrix spectrum Bootstrap | $B = 2000$ | 0.06 <br> 0.05 | 0 <br> 0 | 404.186 <br> 126.859 |
| frequency distortion | $B$-test | $B = 2$ <br> $B = \sqrt{n}$ | 0.058 <br> 0.027 | 0.4180 <br> 0 | 0.0038 <br> 0.1809 |
| | Gram matrix spectrum Bootstrap | $B = 2000$ | 0.05 <br> 0.06 | 0 <br> 0 | 403.391 <br> 122.429 |

Table 1: Synthetic benchmarks.

Figure 1: three synthetic benchmarks.

# References

[1] A Gretton, B Sriperumbudur, D Sejdinovic, H Strathmann, S Balakrishnan, M Pontil, and K Fukumizu. Optimal kernel choice for large-scale two-sample tests. In *Advances in Neural Information Processing Systems 25*, pages 1214–1222, 2012.