[Reviews · NeurIPS 2013]

Submitted by Assigned_Reviewer_2

The authors present an interesting contribution of a block splitted MMD based two-sample test. The authors give a very good and clear motivation in the introduction. It is proposed and described with respect to the two extremes of MMD_u and MMD_l. A recommendation for the block size B is given. Besides the theoretical contribution, the method is convincingly illustrated on several data sets. The work can be promising e.g. towards big data.

On the other hand, with respect to the bootstrap, also block based methods have been proposed:

A scalable bootstrap for massive data. A. Kleiner, A. Talwalkar, P. Sarkar and M. I. Jordan. arXiv:1112.5016, 2012.

It would be good to position the work also with respect to these results.

When reading the abstract it is not immediately clear what the authors mean by B-test. I would suggest to already explain in this early stage that B stands for block splitting. Currently, one expects from the name B-test in the title and abstract that is is a completely new approach, which is not really the case.

typo: Komogorov should be Kolmogorov

When taking sigma = 1, isn't a suitable scaling of the data assumed here? please explain.

Eq 13: shouldn't this be rounded to the nearest integer?

The choice sigma \in {-15, ..., 10} is unclear to me, what do you mean and why negative values? Is this choice crucial or not?

Please explain in more detail what you mean by multiple kernels.


The authors have carefully answered to the questions. I have updated my score.


Summary: Interesting variant of MMD type two-sample test that is applicable to large data sets. Though the connections with the existing literature are very well described, it is a pity that bootstrap for large data sets is not included.

Submitted by Assigned_Reviewer_6

This manuscript takes a well known statistical test, maximum mean discrepancy, and improves its practical value by reducing its computational cost to achieve the same statistical power. Basic idea is to use intermediate sized blocks of samples. This applies to the large sample problems where the exact computation is too slow.

Quality:
The theory as well as the experiments are solid.

Clarity:
I had no problem understanding the details. But the paper assumes the readers are familiar with previous works on MMD.

Originality:
It is a novel combination of previous works that surpass its predecessors.

Significance:
MMD is a powerful nonparametric divergence with high statistical power. Improving its speed has practical importance.

Additional comments:
- The failure of median kernel is interesting, but the authors do not discuss the details.
Summary: Important improvement in a two-sample test widely accepted in NIPS community.

Submitted by Assigned_Reviewer_8

This paper proposes a new MMD kernel two-sample test that is efficient, statistically and computationally. The basic idea is an improvement of the idea behind MMD_l which was suggested in an earlier paper by Gretton et al. While MMD_l uses blocks of size 2, the method of this paper suggests general block sizes, B. The statistical aspects associated with this new test are worked out cleanly using central limit theorem and other well-known results.

The paper is good: the ideas are motivated well, analysis done cleanly and a clean set of experiments are given to show the value of the new method. The only negative aspect is that the contribution is specific and incremental.

The paper recommends B=sqrt(n). Is this done to make the sizes of the two levels, i.e., B and (n/B) equal? It is worth mentioning that the complexity of the test is O(n^1.5).

It would be useful to define sample complexity somewhere before Table 2.

In figure 1 H_0 and H_A are marked incorrectly in the legend.

In the previous to last paragraph of section 4, line numbers 417-418 of the pdf, it is mentioned that the empirical variance is quite different from what the theory predicts. What does that say about the usefulness of theory?
Summary: The paper is done well, but the contribution is specific and incremental.
Author Feedback

Author rebuttal: We thank the reviewers for their helpful feedback and assessment of our work. Our replies are as follows:

Rev. 2
Thank you for drawing our attention to the "Scalable Bootstrap" paper. Briefly, this paper splits the data (of size n) into s blocks, then computes an estimate of the n-fold bootstrap on each block, and averages these estimates. The difference with respect to our method is that we use the asymptotic distribution of the mean of our block statistic to determine a test threshold, whereas the scaleable bootstrap paper is concerned with proving the consistency of a statistic obtained by averaging over bootstrap estimates. We will add this discussion and citation to a final version.

We will explain that B-test refers to block splitting immediately in the abstract.

For \sigma=1: the choice \sigma=1 is indeed an arbitrary default choice (as is the median), and requires the data to be on this lengthscale (which it was). Our intention in including \sigma=1 and the median was to show that a learned kernel outperformed these heuristics.

Regarding the values of the parameter sigma: in writing {-15, ... 10}, we meant to say it was set to {2^-15, 2^-14, ... 2^10}.

By multiple kernels, we mean that every value of sigma defines different kernel over the data points, which are then combined. We considered Gaussian kernel :

k(x, y) = exp(-(x-y)^2 / (2 * sigma))

We used non-negative linear combinations of these kernels, where the coefficients were computed by adapting the approach of [8] to the block case.

Rev. 6: the reason for the failure of the median kernel is that the lengthscale of the difference between the distributions p and q differs from the lengthscale of the main data variation (which is reflected in the median). We will add this point to the present paper to make it self-contained, but for the moment, the reviewer may refer to Section 5 of [8] for more detail.

Rev. 8:

Regarding motivation: we note that degenerate U-statistics with intractable null distributions occur widely in the testing literature, e.g. also in independence testing ("Brownian distance covariance", Szekely and Rizzo, 2009; "A Kernel Statistical Test of Independence", Gretton et al., 2007). A similar pathological null distribution is also found when normalized statistics like the Fisher discriminant are used, as in [11]. In all these cases, our approach may be applied to obtain computationally efficient yet powerful tests without having to deal with a difficult null distribution, or an expensive bootstrap procedure. We will emphasize this broad applicability in our revision. We have prepared downloadable code to make the test easy for practitioners to apply.

We will add the sample complexity. The choice B=sqrt(n) is a heuristic, and just one choice that fulfils our assumptions in lines 265 and 266.

lines 417-418: when p and q are very different, the empirical variance curve matches behavior from equation (7) even at small sample sizes. When p and q are very close, and the number of samples is small, then the asymptotic regime is not reached for the statistic, and eq. 10 becomes a reasonable approximation. As the number of samples increases, and the asymptotic regime is approached, the variance of eq. 7 is again observed to hold.